# Temporal Variability of Virioplankton during a *Gymnodinium catenatum* Algal Bloom

**DOI:** 10.3390/microorganisms8010107

**Published:** 2020-01-12

**Authors:** Xiao-Peng Du, Zhong-Hua Cai, Ping Zuo, Fan-Xu Meng, Jian-Ming Zhu, Jin Zhou

**Affiliations:** 1The Shenzhen International Graduate School, Tsinghua University, Shenzhen 518055, China; 2The School of Geography and Ocean Science, Nanjing University, Nanjing 210000, China; zuoping@nju.edu.cn; 3Second Institute of Oceanography, Ministry of Natural Resources, Hangzhou 310000, China

**Keywords:** algal bloom, phage, metagenomics, viral communities, dinoflagellates

## Abstract

Viruses are key biogeochemical engines in the regulation of the dynamics of phytoplankton. However, there has been little research on viral communities in relation to algal blooms. Using the virMine tool, we analyzed viral information from metagenomic data of field dinoflagellate (*Gymnodinium catenatum*) blooms at different stages. Species identification indicated that phages were the main species. Unifrac analysis showed clear temporal patterns in virioplankton dynamics. The viral community was dominated by *Siphoviridae*, *Podoviridae*, and *Myoviridae* throughout the whole bloom cycle. However, some changes were observed at different phases of the bloom; the relatively abundant *Siphoviridae* and *Myoviridae* dominated at pre-bloom and peak bloom stages, while at the post-bloom stage, the members of *Phycodnaviridae* and *Microviridae* were more abundant. Temperature and nutrients were the main contributors to the dynamic structure of the viral community. Some obvious correlations were found between dominant viral species and host biomass. Functional analysis indicated some functional genes had dramatic response in algal-associated viral assemblages, especially the CAZyme encoding genes. This work expands the existing knowledge of algal-associated viruses by characterizing viral composition and function across a complete algal bloom cycle. Our data provide supporting evidence that viruses participate in dinoflagellate bloom dynamics under natural conditions.

## 1. Introduction

Phytoplankton are important primary producers in marine ecosystems and play a critical role in element cycling, food web production, and broader ecosystem functions [1,2,3]. Over the last several decades, eutrophication, temperature increases due to global climate change, and modifications of the food web have been associated with an increased frequency of phytoplankton blooms with toxic or other deleterious properties. In recent years, worldwide increases have been reported in the occurrence, geographic expansion, and persistence of harmful algal blooms (HABs) [4,5,6], with severe impacts on human health and ecosystems, mainly through the release of toxins and the production of dissolved organic matter [7]. During the early part of the twenty-first century, urbanization has expanded, particularly in coastal and developing countries, making it essential to conduct intensive multidisciplinary research on HABs [5].

In recent years, increases in the frequency and global distribution of HABs has prompted researchers to investigate their triggers and formation mechanisms because it is crucial to understand the factors regulating their dynamics to effectively manage and control them. Biological agents, including plants [8], protozoa [9], bacteria [10], and viruses [11], are considered to be effective and biosecure ways of influencing HAB occurrence and dynamics, and the relationship between microbes and algae has received particular interest in recent years. This partly is due to the roles of microbes in mediating matter cycling, micro-food web structures, and the production of essential elements that stimulate algal growth [12,13,14] as well as in absorbing essential elements [15], exhibiting algicidal activity [16,17], inhibiting sexual reproduction [18], and regulating algae–bacteria signaling (such as quorum sensing) [19]. 

HABs are rapid expansions of phytoplankton populations that represent a major threat to the health of diverse coastal and freshwater aquatic ecosystems [4,5]. HAB formation is significantly influenced by algal-microbial interactions. During bloom events, the structure of the microbial community becomes complex. It is affected by environmental conditions, algal species, their physiological status, and the stage of the bloom [20]. Furthermore, it has been well established that the microecological behaviors of bacteria and archaea create a regulatory network that operates throughout bloom formation, duration, and collapse [21,22]. Viruses are the most abundant biological entities in the sea, and in theory, they can significantly regulate algal dynamics. The shifts in microbial communities during HABs have been widely investigated, but the feature of virus communities during HABs is poorly understood.

Viruses are now indisputably viewed as necessary components of aquatic environments, both in their abundance and their multiple ecological functions [23,24]. They can significantly affect primary production [25], playing a key role in population mortality [26], nutrient cycling [27], and plankton biodiversity [28]. Several studies have demonstrated that viruses are a major contributor to phytoplankton mortality through infection and lysis, and they play an important role in the regulation of algal bloom dynamics [29,30,31]. Tomaru et al. [32,33,34] found that RNA viruses may be particularly relevant to the ecology of phytoplankton blooms. Moreover, viral abundance shows a significantly positive relationship with mainly bacterial hosts, which in turn are correlated with algal biomass [35,36,37]. These results indicate that viral infection is a major factor controlling phytoplankton succession and that viruses are instrumental in the collapse of algal blooms and increase mortality among algae [38,39,40]. 

The important role that viruses have in phytoplankton blooms has been explored in many works on diatoms and blue-green algae. However, few studies have studied the successional trajectories of viral communities with regard to dinoflagellate blooms, especially throughout a complete algal bloom cycle. In addition, despite the relatively mature understanding of viral dynamics during HAB events, the identification of potential functions remains a challenge, limiting our current understanding of the complex interactions between viruses and algae. These considerations indicate the need to study virioplankton dynamics and their potential functions during dinoflagellate blooms in developing our understanding of algal-viral interactions.

Here, using the virMine tool [41], we analyzed the viral information from the metagenomic data of field dinoflagellate (*Gymnodinium catenatum*) samples. We want to test the hypothesis that viral community structure and function are linked to specific bloom stages and co-regulate the fate of HABs. The results may contribute to the current understanding of the viral regulatory role in algal bloom events.

## 2. Materials and Methods

### 2.1. Sample Collection and Environmental Parameters

Experimental samples were collected from a *G. catenatum* bloom in YanTian (22°35′00″ N, 114°16′00″ E), Shenzhen, China. Sampling was carried out in the field from 4 August 2015 to 10 September 2015 (eleven sampling time points; referred to as YT1 to YT11). Water samples were collected in triplicate (5 L each) using a Niskin bottle, from depths of approximately 0.5–1.0 m. Samples were brought to the laboratory under ice-bathing condition for further treatment. A 50 mL fraction of each sample was preserved in 1% glutaraldehyde to quantify total bacterial abundance. In addition, 50 mL of each sample was preserved with 1% Lugol’s iodine for the identification and quantification of phytoplankton. Algae were observed and counted under an optical microscope (×100 magnification).

In addition to the evaluation of each sample, physicochemical parameters of the seawater (temperature, salinity, pH value, and Chl α level) were recorded in vivo using a YSI 6920 water-quality sensor (Xylem Inc., Yellow Springs, OH, USA). Other environmental factors, including levels of ammonium nitrogen (NH_4_^+^), nitrate nitrogen (NO_3_^–^), nitrite nitrogen (NO_2_^–^), and phosphate phosphorus (PO_4_^3–^), were measured according to the methods of Murphy [42] and Greenberg et al. [43].

### 2.2. Collection of Environmental DNA

The seawater samples were initially passed through 10 μm filter to remove particles or cells. For enrichment of marine viruses, we used FeCl_3_-mediated flocculation, as described previously [44]. In brief, we added 6 mL of 1% aqueous FeCl_3_ into 60 mL of seawater and mixed it thoroughly. The sample was then incubated at room temperature for at least 1 h. The incubated seawater was then passed through a polycarbonate membrane filter (0.22 μm, Millipore, Billerica, MA, USA) and stored at 4 °C until analysis. The filtered membranes, including viral particles and microbial cells, were collected for DNA extraction. Total DNA was extracted using the MoBio PowerWater Viral DNA/RNA Isolation Kit (MoBio laboratories, Carlsbad, CA, USA) according to the manufacturer’s instructions. The viral DNA was amplified by Qiagen REPLI-g Single Mini Kit (Qiagen, Hilden, Germany) using Phi29 DNA polymerase. The viral clone library was constructed using the whole-genome multiple displacement amplification (MDA) method. The sequencing library for next-generation sequencing (NGS) was prepared using NEBNext^®^ UltraTM DNA Library Prep Kit (New England Biolabs, Ipswich, MA, USA). Adapter-ligated fragments were then amplified using PCR and purified by gel electrophoresis. The DNA library was pair-end sequenced using Illumina HiSeq 2000 by MAGIGENE Biotech. Co. Ltd. (Guangzhou, China).

### 2.3. Metagenomic Sequencing

Among the 11 time-point samples obtained during the HAB event, we selected 8 for metagenomic analyses: YT1, YT3 (pre-bloom), YT4-YT6 (during bloom), and YT8, YT10-YT11 (post-bloom). DNA extraction was performed using the aforementioned methods. Purified genomic DNA was sheared to 150–200 bp using the Covaris S220 (Woburn, MA, USA) acoustic system. Metagenomic libraries were prepared using the NuGen Encore multiplex systems I and IB (NuGen, Carlsbad, CA, USA). The manufacturer’s instructions were followed for end repair, adapter ligation, amplification, and barcoding, with unique DNA barcodes flanking the adapters for each sample. The metagenomes were sequenced using the next-generation sequencing platform Illumina HiSeq 2000 (MAGIGENE Biotech. Co. Ltd., Guangzhou, China).

### 2.4. Data Analysis

For environmental parameters, SPSS version 13.0 was used to perform statistical analysis. Then, one-way analysis of variance tests was conducted to evaluate the differences between different HAB stages, and adjusted *p*-values (0.05 or 0.01) were determined. For viral composition, the cluster and relationships between viral community composition and environmental parameters were investigated using correlation matrices (such as Pearson correlation) and Euclidean distances under R software (www.r-project.org). Inter-species associations between virus and target species (*G. catenatum*) were examined using a variance ratio (VR) test [45]. To visualize the results, a network was created to identify inter-species associations and correlations among species [46].

Raw sequencing reads underwent quality trimming using Trimmomatic to remove adaptor contaminants and low-quality reads [47]. Read counts were normalized according to Yang et al. [48], dividing the raw-read count by the total number of reads in the sample and by the (average) genome size. The abundance of each gene within a sample was calculated as follows:(1)$$a_{i}=\frac{b_{i}}{\sum_{j}b_{j}}=\frac{\frac{x_{i}}{L_{i}}}{\sum_{j}\frac{x_{j}}{L_{j}}}$$
where *a_i_* represents relative abundance of (*i*) gene; *L_i_* represents the length of (*i*) gene sequence; *x_i_* represents the number of (*i*) gene-mapping reads; *b_i_* represents the copy number of (*i*) gene; and *j* represents the number of all of reads. Uncontaminated high-quality reads (clean reads) were used for subsequent analysis. Clean reads were then independently assembled to scaffolds for each sampling site using SPAdes [49]. The resulting scaffolds were merged, and redundancy was removed by CD-hits at 90% identity. MaxBin 2.0 (http://downloads.jbei.org/data/microbial communities/MaxBin/MaxBin.html, Version 2.2.4) was used to perform scaffold binning by adopting the following conditions: (1) assembled scaffolds had lengths >1000 bp; (2) tetranucleotide frequencies and coverage level of scaffolds were calculated; (3) the parameters were substituted into the expectation maximization (EM) algorithm to calculate the probability that all scaffolds belong to each bin. Subsequently, the contigs were uploaded to virMine to perform automated analysis and annotation. Open reading frames (ORFs) were predicted from the scaffolds. To investigate the functional diversity of HABs viruses, clusters of orthologous group (COG) annotation of viromes was determined by Blastp comparisons of predicted ORFs with the eggNOG database (http://eggnog.embl.de/) with an E-value of 1 × 10^−5^. ORFs affiliated to COG function classes of carbohydrate transport and metabolism were further clustered to generate unique carbohydrate metabolic ORFs with CD-hits. Subsequently, CAZymes from these viral ORFs were identified on the dbCAN web server based on CAZyme family-specific HMMs [50]. ORFs related to carbohydrate metabolism and CAZymes were compared with NCBI nr and Pfam databases to determine the most accurate annotation and similarity for each ORF. For map analysis, scaffolds containing CAZymes were retrieved, and ORFs were identified with Meta-Gene. The ORFs were then compared with NCBI for functional and taxonomic annotation based on protein sequences.

### 2.5. Data Availability

The raw data in this study have been deposited in the NCBI database under accession number SRP158562. In addition, the analyzed data of viral scaffolds can be found in Appendix A.

## 3. Results and Discussion

### 3.1. Algal Bloom Characteristics

During this algal bloom, the target species was *G. catenatum*, and cell concentrations ranged from (0.51 ± 0.042) × 10^2^ to (5.4 ± 0.61) × 10^3^ cells/mL, with the highest biomass appearing at the peak stage (*p* < 0.01). Five phases were documented over the course of the bloom’s cycle: pre-stage (YT1 and YT2), exponential growth stage (YT3 and YT4), peak-stage (YT5 and YT6), decline-stage (YT7–YT9), and terminal-stage (YT10 and YT11). During the HAB sampling period, temperature, salinity, and pH ranges were 27.5–28.7 °C, 27.8%–34.6%, and 7.36–8.52, respectively. There was distinct temporal heterogeneity in the main nutrient concentrations (NO_3_^–^, NO_2_^–^, NH_4_^+^, and PO_4_^3–^) throughout the HAB, and the lowest values appeared at the bloom’s peak. Detailed information is given in our previous work [51]. The changes in these parameters reflect the basic ecological characteristics of the algal bloom, indicating the typical dynamic characteristics of dinoflagellates [52].

### 3.2. Metagenomic Data

We got 242.5 gigabases of raw reads (mean of 31,051,745 per sample) from the eight metagenomic samples. Details of the data can be found in a previous work [51]. After primary assembly, a total of 15,472,449 scaffolds (>200 bp) were obtained in the test samples. We selected the scaffolds (>1000 bp) that were predicted as viral and queried them via megablast against the NCBI nr/nt database online. A total of 428,502 scaffolds were compared to the nt database, resulting in the identification of 25,263 viral scaffolds, which were taxonomically assigned as viral sequences. 

### 3.3. Taxonomic Composition

The most abundant viral communities were double-stranded DNA (dsDNA) viruses (73.08%), with a relatively lower proportion (<1%) of ssDNA and RNA virus communities. Most of the dsDNA groups of viruses were bacteriophages, which reflects the high abundance and diversity of bacterial hosts during the algal bloom. In addition, the most dominant dsDNA viruses were from the order *Caudovirales*. Previous viral metagenome studies on marine environments have also found a high abundance of *Caudovirales* [53,54]. Our analysis indicated that the relative abundance of *Caudovirales* increased from the pre-bloom stage (YT1 and YT3) to the growth and peak stages (YT4 to YT6); it reached its highest values at YT5 (50.6%) and subsequently decreased during the post-bloom stage (YT8, YT10, and YT11) (Figure 1). This trend suggests that the relative abundance of *Caudovirales* has distinct alterations in different algal bloom stages. In addition, *Caudovirales* has been reported to infect a wide range of bacterial hosts, especially species in the Proteobacteria and Bacteroidetes phyla, which are dominant taxa in blooms [51]. Bacterioplankton are important microbial components of food-web fluxes [39], and they have been associated with natural plankton populations [55]. Therefore, it is possible that *Caudovirales* potentially or indirectly affects the structure of the plankton community during blooms.

At the family level, the overall viral community was primarily represented by *Siphoviridae*, *Podoviridae*, *Myoviridae*, *Phycodnaviridae*, *Mimiviridae*, and *Microviridae* (Figure 2). As the bloom developed, significant changes were observed in the viral community. The relative abundance of *Podoviridae* showed a declining trend, from 32.8% ± 0.6% in the pre-bloom stage to 15.1% ± 7.6% in the peak stage, followed by a mild increase during the post-bloom stage (21.4% ± 5.6%). The patterns observed among the *Myoviridae* and *Siphoviridae* were somewhat different, with a gradual increase over time and a maximum proportional abundance appearing at the peak stage, with approximately 21.9% ± 11.1% and 50.5% ± 24.6% of the total OTU tags, respectively. Similarly, the abundance of the other members (such as *Phycodnaviridae* and *Microviridae*) showed a dramatic increase during the *G. catenatum* bloom and were enriched (with abundances of more than 10%) at the end of bloom. *Siphoviridae*, *Podoviridae*, and *Myoviridae* were the major members, which is not surprising because these species have been found to be the most abundant groups of viruses in the aquatic environment [54,56,57]. Previous metagenome investigations of environmental samples have shown ubiquitous dsDNA eukaryotic viruses (such as *Iridoviridae*, *Phycodnaviridae*, *Mimiviridae*, *Poxviridae*, and *Polydnaviridae*) in the DNA viromes. In our study, only *Phycodnaviridae* and *Mimiviridae* were detected, and they accounted for a relatively low proportion of the whole. The main reason for this was likely our choice of sampling sites: Gong et al. [54] used samples from deep sea areas, whereas our samples were collected near shore in a phycosphere environment. The relatively higher abundance of *Phycodnaviridae* and *Mimiviridae* in this work indicates that they were the main species associated with *G. catenatum* and perhaps participated in the decline of algal bloom [58,59,60,61]. In addition, we identified an extremely small fraction (0%–3.0%) of *Microviridae*, as the sole ssDNA viruses. This low level of representation may reflect differences in the abundances of these viruses across bloom stages. However, many new ssDNA viral genomes have been assembled from environmental samples from Saanich Inlet, the Strait of Georgia, and the Gulf of Mexico [62]. This suggests that there exist a great number of ssDNA viruses in the ocean that are as yet unrecognized [63]. In future work, researchers should improve the methods of amplification and increase sequencing depths to mine information on ssDNA viruses in algal bloom environments.

Further investigation at the genus level indicated that 41 common viral species were present in the entire sample, including 31 species of phages (Figure 3). The top six most abundant species were uncultured *Mediterranean* phage uvMED, *Celeribacter* phage P12053L, *Flavobacterium* phage FL-1, uncultured *Mediterranean* phage uvDeep CGR1-KM17-C101, *Roseobacter* phage SIO1, and *Verrucomicrobia* phage P8625. In a previous study, Hwang et al. [56] found that *Pelagibacter* phage HTVC010P, *Ostreococcus lucimarinus* OIV5/OIV1, and *Roseobacter* phage SIO1 were the most common species in Goseong Bay. Gong et al. [54] suggested that *Cellulophaga* phages, *Pseudomonas* phages, and *Vibrio* phages were responsible for the high percentage of surface water viromes in Prydz Bay. In Chile Bay, an integrative omics study demonstrated that the phage community associated with bacterioplankton was dominated by a new *Pseudoalteromonas* virus (*PpCBA*) during a diatom-dominated phytoplankton bloom [64]. As the main driver, bacteria play multiple ecological roles in phytoplankton blooms, killing hosts, degrading biopolymers, and recycling algae-derived organic matter. The high abundance of phages in the phycospheric environment may act as a potential factor to affect bacterial composition and eventually contributes to the initiation and termination of algal blooms.

### 3.4. Associations between Viral Communities and Environmental Variables

Figure 4 shows the associations between viral composition and environmental parameters. Among the physical factors, the dominant member was temperature, which indicates that temperature is a major abiotic force that shapes the dynamics of viral community dynamics. Among the chemical factors, multiple inorganic nutrients (NO_2_^−^, NO_3_^−^, NH_4_^+^, and PO_4_^3−^) co-contributed to the variability of viral communities. Specifically, *Roseobacter* phage SIO1, *Flavobacterium* phage FL-1, and *Verrucomicrobia* phage P8625 were positively correlated with NO_2_^−^ and NO_3_^−^ and were negatively correlated with PO_4_^3−^; *Prochlorococcus* phage Syn1 and *Pseudoalteromonas* phage BS5 were positively correlated with PO_4_^3−^ and NO_2_^−^ to some extent, but NH_4_^+^ had a negative influence; and in uncultured Mediterranean phage members, most genera exhibited positive associations with nutrients. Our results are consistent with Deng et al. [65], and demonstrate that the nutrient-mediated succession of the virioplankton community contributes to bottom-up viral composition. Teeling et al. [66] found that material resources trigger succession of microbial populations; based on this, we speculated that community succession (like virus) in algal blooms is also regulated by substrate availability. However, more ecological data will be needed to confirm this in future work.

Although abiotic factors generally control viral communities, biotic correlations may also play a role. In addition, in order to investigate the contribution of viral interactions to the host, we constructed a network based on correlations between the predominant taxa and the target species, *G. catenatum* (Figure 5). At the genus level, there were forty-four correlations (twenty-seven negative and fourteen positive) between the viral phenotypes and target algae (*G. catenatum*). *G. catenatum* was strongly positively correlated with *Pelagibacter* phage HTVC008M, *Flavobacterium* phage FL1, and *Bathycoccus* sp. RCC1105 viruses in the network (*p* < 0.01), and negatively correlated with *Synechococcus* phage S-EIV1, uncultured Mediterranean phage uvDeep-CGR1-KM17-C101, and CGR2-KM23-C896, among others (*p* < 0.01). This multiplicity of positive inter-species associations may have provided an opportunity for species to adapt to the surrounding environment and contribute to the duration of algal bloom [67]. By contrast, the negative correlations reflect the appearance of kill or lyse algal cells. Lysis activity related to bloom-forming species, such as *Emiliania huxleyi*, *Phaeocystis pouchetii*, and *Phaeocystis globose*, has been well studied [38,64]. During such blooms, viruses exhibit a strong regulation activity and contribute to HAB collapse in a boom-and-bust pattern [68,69,70]. Similarly, we observed a decline in *G. catenatum* at the post-HAB stage, which can be explained with the “killing the winner” hypothesis [71], wherein viral lysis of the most abundant algae releases new nutrients and opens a niche that other algal species then compete for.

### 3.5. Functional Prediction of Algal Bloom Sample Viromes

Given many viral species are uncultured in the laboratory, the reference genome database for viral metagenomic analysis is still far from complete, and an extraordinary amount of uncharacterized viral “dark matter” continues to exist in seawater. For this reason, there are many sequences that remain without an assigned function [72,73,74]. Most recognizable functions were related to conventional viral functions, such as replication, recombination and repair, intracellular trafficking, secretion, vesicular transport, biogenesis of cell wall/envelope/membrane, DNA/RNA transport and metabolism, as well as transcription (Figure 6a), which are critical for viral reproduction and survival. According to the annotation results from the CAZy database, the ORFs of 43 auxiliary CAZyme genes are mostly affiliated with glycoside hydrolase (GH) and glycosyl transferase (GT) (Figure 6b), implying the importance of these functional genes for the degradation of marine organic carbons. Actually, complex carbohydrates, such as starch, xylan, cellulose, chitin, alginate, mannan, and pectin, are major components of algae cell walls and microbial intercellular spaces, and they are highly difficult to degrade [75]. Notably, in viruses, multiple genes encode CAZymes, such as hydrolysis enzymes and auxiliary enzymes, which are necessary for the degradation of various extracellular polymeric substance (EPSs) [76], indicating viral abilities in the biolysis of complex polysaccharides during bloom events. In this study, we found most viral carbohydrate metabolic genes belonged to CAZymes with glycoside hydrolase activities, which allow viruses to primarily participate in the decomposition of organic carbon during algal bloom events. To better elucidate the distribution of CAZymes at different bloom stages, the relative abundance of GH and GT was investigated (Figure 6c). The relative abundance of GH significantly increased (from YT01 to YT04), reaching peak abundance (>75%) at YT04, and then gradually increased from the later during-bloom stages to the post-stages. The trend of the relative abundance of GT was opposite to that of algal bloom formation and collapse. It seems that this big variation in the relative abundances of GH and GT was associated with bloom dynamics. In order to identify the potential contributions to carbon degradation, the relative contributions of viral species to GH and GT genes were performed. Figure 6d shows that genes encoding components of the GH enzyme were mostly affiliated with the uncultured *Mediterranean* phage, *Celeribacter* phage, *Flavobacterium* phage, and *Roseobacter* phage; whereas genes in the GT proteins mostly belonged to uncultured *Mediterranean* phage, *Verrucomicrobia* phage P8625, *Vibrio* virus nt1, and uncultured marine virus (Figure 6d). In particular, the uncultured *Mediterranean* phages co-contributed to GH and GT genes, suggesting that these viruses are main contributors to organic carbon decomposition [77]. Viral “auxiliary metabolic genes” (AMGs) for carbon metabolisms have been extensively investigated in marine environments; most of them are involved in central carbon metabolism to facilitate viral replication [78]. In this work, we found various viruses associated with GH/GT genes, and we speculated that virus-encoded GHs/GTs can act as AMGs and increase host accessibility through producing more viruses [79]. Recent work proposed that virus-encoded GHs potentially augment the breakdown of complex carbohydrates to increase energy production and boost host metabolism during viral infection [80,81]. These results hint that viruses promote host infection and induce HAB decline. 

## 4. Conclusions

This study surveyed viral communities in a natural dinoflagellate *G. catenatum* bloom and found that the viruses exhibited remarkable heterogeneity in their responses to algal bloom events. Viral community structures were affected by multiple factors, including physiochemical parameters, substance availability, and interactions. A multitude of viral interactions (positive and negative) are likely involved in determining a given algal bloom fate at a specific time and location. Functional predictions indicated that the viruses exhibited glycoside hydrolase activities, facilitating the understanding of the primary roles of viruses in the degradation of organic carbon. Detailed comparisons of the metabolic characters of phytoplankton and viral communities should be performed to enable a better understanding of the ecological mechanisms of HABs from a virological perspective.

## Figures and Tables

**Figure 1 microorganisms-08-00107-f001:**
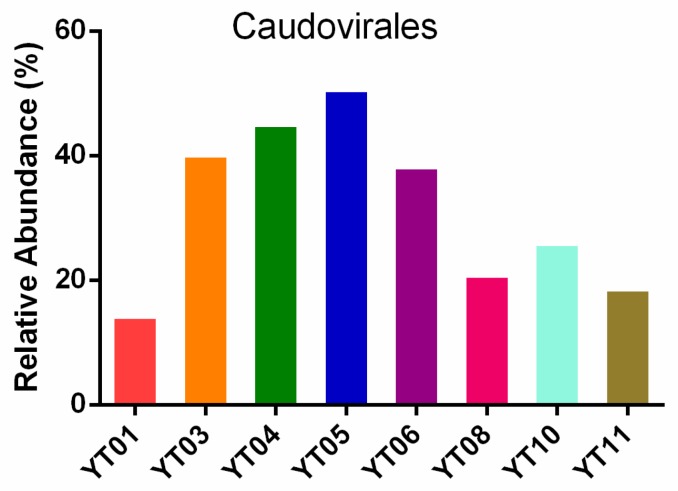
Relative abundance of Caudovirales in different algal blooming samples.

**Figure 2 microorganisms-08-00107-f002:**
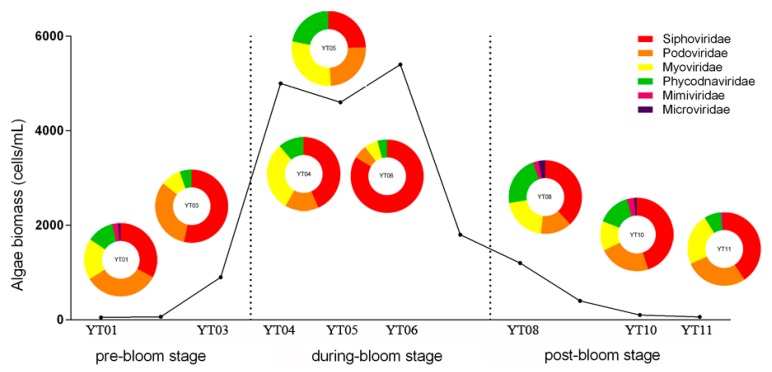
Taxonomic compositions of viral communities at the family level in different bloom stages. The pie charts show percent relative abundances of viral groups for all eight samples.

**Figure 3 microorganisms-08-00107-f003:**
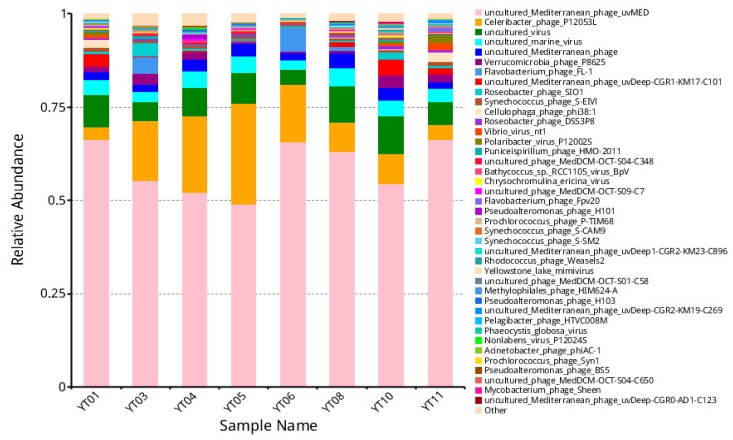
Relative abundance of the 41 most common viral species in *G. catenatum* bloom samples.

**Figure 4 microorganisms-08-00107-f004:**
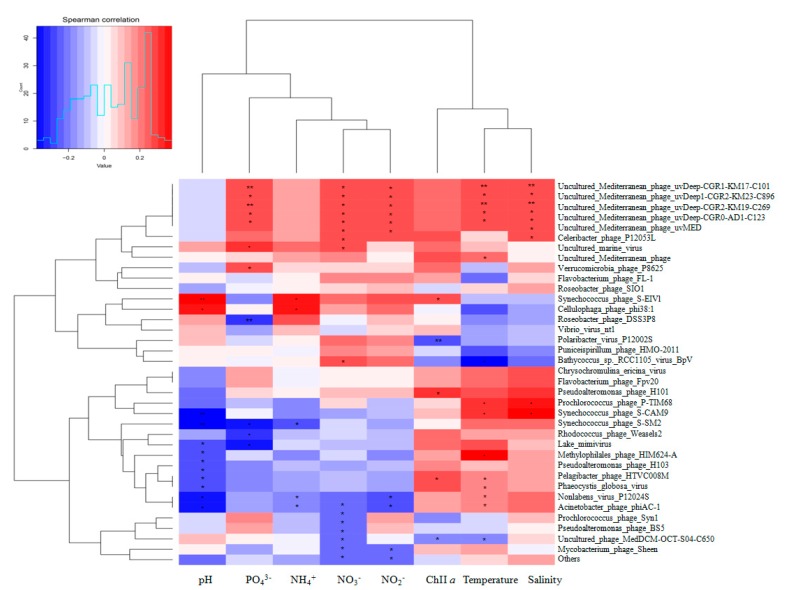
Correlation analysis between relative abundances of main viral species and environmental parameters based on Pearson correlations. The r-values indicate the correlation values of Pearson correlations. The P values indicate the statistical significance levels (single* *p* < 0.05 and double ** *p* < 0.01).

**Figure 5 microorganisms-08-00107-f005:**
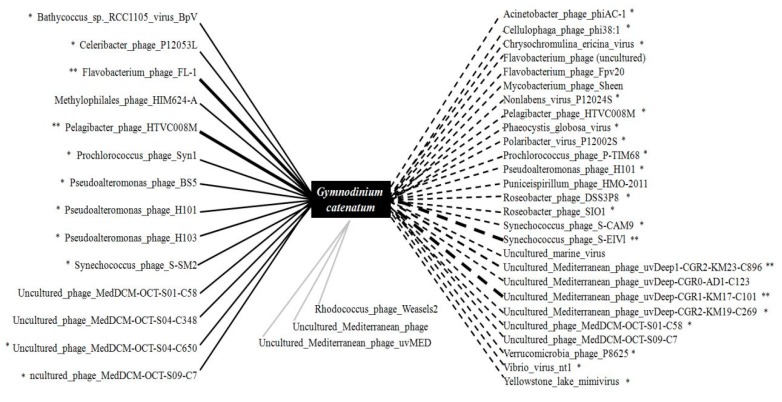
Correlation analysis of the relationship between the representative virus and the target algae *G. catenatum*. There was a total of forty-four correlations (fourteen positive, twenty-seven negative, and three unconnected). The solid lines indicate positive correlations, and bold solid lines indicate strong, negative correlations. The dotted lines indicate negative associations, and the bold dotted lines indicate strong negative associations. The grey solid lines indicate unlinked relationships. The statistical significance levels are * *p* < 0.05 and ** *p* < 0.01.

**Figure 6 microorganisms-08-00107-f006:**
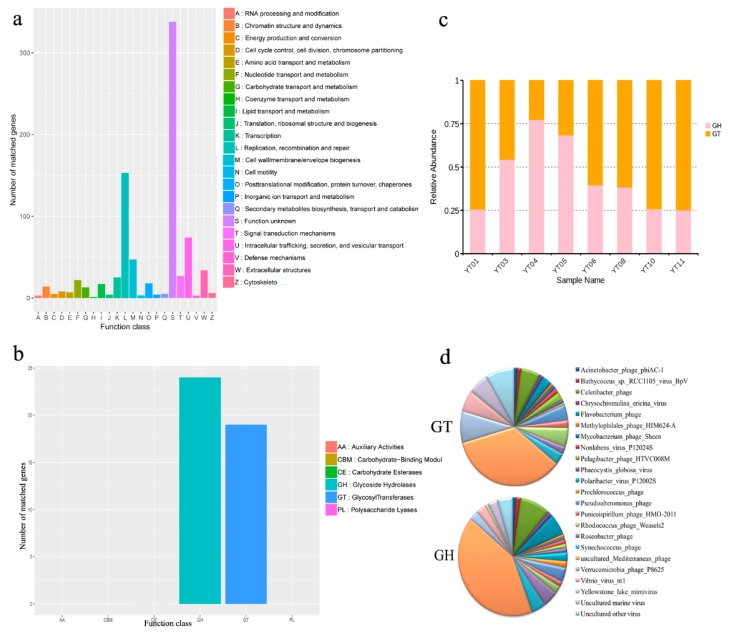
Overview of functional annotation algal bloom sample viromes. (**a**) Composition of predicted functional genes of sample viromes. (**b**) Annotation of viral carbohydrate-metabolism-related ORFs in the CAZy database. (**c**) Relative abundance of GH and GT in different algal bloom samples. (**d**) The relative contributions of viral species to GH and GT genes.

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
