# Peer review of "Temporal Variability of Virioplankton during a Gymnodinium catenatum Algal Bloom"

_microorganisms, 2020, doi:10.3390/microorganisms8010107_

Round 1

Reviewer 1 Report

This manuscript describes the temporal variability of viruses during a dinoflagellate bloom.  Probably not surprisingly the majority of the viruses were bacteriophage and the community structure changed during the bloom formation and disappearance.

I only have a few editorial comments.

lines 87 to 88.  You state that you took weekly samples over an ~5 week period and you had 11 sampling points.  Something is not quite right.

line 90. Samples were brought to the....

line 93. Algae were...

line 184. It does not look like the Mimiviruses make up 10% of the population - see fig. 2.

It is very hard to see the legends on some of the figures - especially Fig. 6.

Author Response

Dear Reviewer 1:

Thank you for your good comments and valuable suggestions concerning our manuscript. Those comments are all valuable and very helpful for revising and improving our article, as well as the important guiding significance to our researches. In the revised manuscript, we accept most of your suggestions and modified the text according your comments. The detailed responses as below:

Point 1: lines 87 to 88.  You state that you took weekly samples over an ~5 week period and you had 11 sampling points.  Something is not quite right.

Response 1: For this question, please allow us to explain. During this work, we carried out a track sampling, the whole cycle is five weeks. In the first two weeks, we collected samples once a week. When the algal bloom occurrence, we collected the samples every day. In addition, we collected the last samples at the end of the algal bloom. The total number of sampling points are 11. The detailed information can be seen from our previous work (Microbiol. Res. 2018, 217, 1-13).

Point 2:line 90. Samples were brought to the....

Response 2: We have revised the sentence like this: "Samples were brought to the laboratory under ice-bathing condition for further treatment".

Point 3:line 93. Algae were...

Response 3:We have revised the sentence as follows: "Algae were observed and counted under an optical microscope (×100 magnification)".

Point 4:line 184. It does not look like the Mimiviruses make up 10% of the population-see fig. 2.

Response 4:Thank you for this meaningful suggestion. After checking, we have confirmed that Mimiviruses didn’t make up 10% of the population. We have revised the sentence as follows: Similarly, the abundance of the other members (such as PhycodnaviridaeandMicroviridae) showed a dramatic increase during the G. catenatum bloom and were enriched (with abundances of more than 10%).

Point 5:It is very hard to see the legends on some of the figures - especially Fig. 6.

Response 5:We checked the entire manuscript and modified some figures, especially the Figure 6. We also replaced  with  new figure in the revised manuscript.

Reviewer 2 Report

In the manuscript titled “Temporal Variability of Virioplankton during a Marine Dinoflagellate Bloom”, the authors presented their analyzed of viral metagenomic sequencing data from 8 samples associated with a dinoflagellate bloom (Gymnodinium catenatum). Viral identification was performed using virMine, and resulting viral taxonomic distribution was discussed in relation to biotic and abiotic trends during the algal bloom. The longitudinal metagenomic dataset of an algal bloom that the authors present is of interest to the field of environmental microbiology. Given the amount of information acquired from deep sequencing and the existence of prior analyses (ref. 45), the authors only performed a cursory analysis in the current manuscript. Detailed methods related to how analyses were performed were missing. In addition, many conclusions stated in the manuscript require more support. Therefore, I do not believe the manuscript is ready to be published in its current form.

Major Comments

I find that methods used to generate figures 4, 5, and 6 are missing from the method section. There is also a lack of discussion regarding what statistic tests are used to perform the correlation analysis and how the p-values are computed. Without such information, it is difficult to judge the confidence associated with the results. In addition, the manuscript its current form is missing a data accessibility section, where links to the raw data, analyzed data, and code used to generate the figures should be included.

Several mechanistic conclusions and claims made in the manuscript are based either on relative abundance or correlations. These claims should either be supported with more data or removed. In line 193-195, the statement “The relatively higher abundance of Phycodnaviridae and Miniviridae in this work indicates that they were the main ‘workers’ that infected or lysed algae and contributed to algal bloom decline” is inferring causation from correlation. More evidence is required to support this conclusion. Similarly, in line 218-220, “the high abundance of phages … indicates that they induce some switch …” is another sentence that seems to be making a mechanistic conclusion from purely relative abundance information. Then, in lines 235-237, “out results support … is regulated by substrate availability” is a sentence that claims a mechanistic relationship based on correlation between relative abundance and abiotic factor measurements. Finally, in lines 254-255, “These results indicate that viruses could be regulators of host biomass …” is similarly not support by evidence presented in the manuscript.

In Section 3.5, I’m confused as to why the authors treated all the annotated genes without considering the contigs they were on. Since the authors are trying to make claims about what the viral genes are and what metabolic processes they participate in, it makes more sense to categorize genes by the genomes they are extracted from (viral, bacterial, or algal). Then, the results would show definitively if viral genomes recovered from the assembly efforts in the prior sections of the manuscript carried GH genes. However, even with such evidence, I feel it is still not enough to make claims related to whether or now the genes are actually used by viruses, which would require more biological evidence.

The only bioinformatic tool used in the manuscript to identify viral contigs is virMine. The currently practice in the field of viromics is to use multiple viral identification tools that may be based on different computational frameworks such as marker gene identification, k-mer frequency, or machine learning. Authors of the current work should also consider using VirSorter, DeepVirFinder, and/or the newly published PhaMers to acquire a more comprehensive view of the viral landscape. In addition, identification of viral contigs is not a straight forward protocol. The authors should discuss confidence associated with each identified viral groups.

Minor Comments

Line 57 – Viruses are not considered microbes. The statement should be corrected.

Line 150-155 – 89,358 contigs (>500bp) were assembled from the 8 metagenomic samples and were described in a previous publication (ref. 45). However, after passing reads from the same set of samples through virMine, 428,502 sequences were obtained. I’m not sure if sequences here mean contigs or reads. The virMine paper seem to suggest that it takes as input contigs, which is confusing since it produced 5 times the number of contigs as previously described. I would like to see the authors describe their process of viral contig identification with more detail, including any cutoffs used to prevent the analysis of contigs that are too short.

Line 107 – Do the authors mean MoBio’s PowerSoil kit?

Line 160 – My understanding of how the DNA isolation kit works is that it effectively removes RNA. The TruSeq library preparation method, on the other hand, does not ligate adapters onto ssDNA or circular DNA strands. Could the authors discuss the mechanism they think resulted in the recovery of ssDNA and RNA viruses?

Line 179 – “… showed a obvious decline” should be “… showed an obvious decline”. In addition, using words such as “obvious” makes the sentences less scientifically rigorous.

Line 246 – Should “inter-specific associations” be “inter-species associations”?

Figure 3 – How are relative abundances normalized? This should be discussed in the method section. In addition, since viral databases are far from complete (which the authors correctly state), what is the justification of normalizing identified viral sequences and discuss their relative abundance as though they represent the entire viral diversity?

Round 2

Reviewer 2 Report

The authors have addressed many of the concerns and have provided more detailed method descriptions. However, some of the points were ignored. More specifically, I would prefer to see the following for other readers to be able to evaluate the conclusions of the manuscript.

A data accessibility section, where links to the raw sequencing data, analyzed data, and code used to generate the figures are included. Even though raw data used for this manuscript is previously published, I would still like to see the SRA or GenBank link included in this manuscript.

The equation and captions as part of “Response 11” included in the Materials and Methods section, even if it is also used in the analysis of ref. 46.

In response to my Point 11, the authors noted that assembled contigs were binned into viral species. Based on my calculation, 25,263 viral scaffolds, 5.9% belong to viral sequences generates ~1500 viral species. In figure 3, what I would like to see is a plot summarizing statistics of those viral genomes. This could be represented as histograms of viral genome sizes as a function of temporal samples.

Are p-values adjusted for multiple hypothesis testing? If so, it should be stated in figure captions along with N, the number of observations adjusted for.

Point 3 has not been addressed fully. Section 3.5 discusses genes found from viral species. More specifically, the authors focus on GH and GT genes. However, there is no discussion regarding which viral species contain these GH and/or GT genes. Since the data and the analyses are already available, it seems feasible to generate an illustration showing which viral species, as a function of time, possesses GH and/or GT genes. This can be done as part of Figure 6 and can be discussed in Section 3.5. It would help address the question “is the trend we see in 6c due to increased GH containing viral species or increased abundance of the same GH containing species”.

The manuscript was well written. However, the edits introduced incorporated more grammatical errors that need to be corrected.

Round 3

Reviewer 2 Report

Based on the added Data Availability Section, I am still unable to find SAMN13475448 to SAMN13475454 in NCBI’s BioSample repository or anywhere else. I feel that in the spirit of disclosing scientific discovery, the manuscript does not warrant publication if raw data is not accessible to reviewers.

After repeated suggestions to include viral contig associated statistics along with Figure 3, the authors seem to be reluctant to provide such details. The authors’ reason is excessive effort, which should not be the case considering there must already be a spreadsheet or database containing information regarding each viral contig. Disclosing such information is important not only for reproducibility purposes but also for readers to evaluate quality of the sequencing data and bioinformatic analyses. The unwillingness to incorporate suggested improvements to the manuscript, especially those related to figures, makes it difficult for me to recommend the publication of this work.

Author Response

  For these questions, please allow us to explain. After receiving the second round of modification notice, we started to revise the manuscript. In the last job, due to our carelessness, we provided the wrong serial numbers (these are the receipt notice numbers for another data in our work). As a result, it prevented you from searching for information in the database. In fact, we have a deposit number in NCBI, and the correct accession number is SRP158562. We have updated the correct accession number (SRP158562) in the revised manuscript. In addition, the data of viral scaffolds (Supplementary File 1) was uploaded as supplementary material in the revised manuscript, and some proper descriptions about the data are also added accordingly (line 165-166).
